# Human Sperm Morphology as a Marker of Its Nuclear Quality and Epigenetic Pattern

**DOI:** 10.3390/cells11111788

**Published:** 2022-05-30

**Authors:** Marion Bendayan, Liliana Caceres, Emine Saïs, Nelly Swierkowski-Blanchard, Laura Alter, Amélie Bonnet-Garnier, Florence Boitrelle

**Affiliations:** 1Reproductive Biology, Fertility Preservation, Andrology, CECOS, Poissy Hospital, 78300 Poissy, France; marion@bendayan.eu (M.B.); lialcaceres@gmail.com (L.C.); sais.emine@gmail.com (E.S.); laura.alter@ght-yvelinesnord.fr (L.A.); 2BREED, INRAE, Paris Saclay University, UVSQ, 78350 Jouy-en-Josas, France; amelie.bonnet-garnier@inrae.fr; 3Reproductive Medicine, Poissy Hospital, 78300 Poissy, France; nelly.swierkowskiblanchard@ght-yvelinesnord.fr

**Keywords:** sperm vacuoles, vacuole, epigenetic mark, histone, chromatin, human sperm, high magnification microscopy, H3, H3K4me3, H3K27me3, embryo

## Abstract

Background: Human sperm chromatin condensation is a sum of epigenetic events that allows for the near-complete replacement of histones with protamines. Under high-magnification microscopy, nuclear vacuoles have been described as thumbprints with poor chromatin condensation. The objective of this study is to examine whether vacuolated spermatozoa carry specific epigenetic marks, which may influence embryo development. Methods: The presence and three-dimensional distribution of ten epigenetic marks (protamine-P2, histone-H3, H3K4me1/me2/me3, H3K9me1/me2/me3, H3K27me3, H4k20me2) were evaluated and compared in morphometrically normal spermatozoa according to the presence or absence of a large vacuole occupying more than 15% of the head surface (n = 4193). Results: Vacuolated spermatozoa were significantly more frequently labelled with H3 and H3K4me3 than normal spermatozoa (88.1% ± 2.7 and 78.5% ± 5.2 vs. 74.8% ± 4.8 and 49.1% ± 7.4, respectively; *p* = 0.009 and *p* < 0.001) and significantly less marked by P2 and H3K27me3 (50.2% ± 6.2 and 63.9% ± 6.3 vs. 82.1% ± 4.4 and 73.6% ± 5.1, respectively; *p* < 0.001 and *p* = 0.028). In three dimensions, vacuoles are nuclear concavities filled with DNA carrying the H3K4me3 marker. Conclusion: High-magnification microscopy is a simple tool to estimate in real time the sperm epigenetic profile. The selection of normal spermatozoa without vacuoles and the deselection of spermatozoa with vacuoles appear to be epigenetically favorable to embryo development and safe offspring.

## 1. Introduction

Sperm nuclear maturation begins after meiosis, during spermiogenesis (the final stage of spermatogenesis, which results in spermatid differentiation into spermatozoa and includes the establishment of the acrosome, the cytoplasm removal, and the condensation of haploid deoxyribonucleic acid (DNA)). This nuclear maturation will continue in the epididymis [1,2]. The structural and molecular changes undergone by the sperm genetic material result in a compact and insoluble chromatin, which protects the sperm DNA during its transport in the genital tract. In mammals, and in humans in particular, histones are initially replaced by transition proteins and then by protamines [3,4,5]. Thus, in human sperm, DNA is mainly linked to protamines (85–90%) and, to a lesser extent, to histones (10–15%) [6,7]. The process of chromatin condensation, which starts in the testes, finalizes during epididymal maturation with the formation of di-sulfide bonds, which further compact the DNA (around a 10-fold increase in chromatin compaction). This gives the sperm chromatin a toroidal structure, also referred to as a donut structure [8,9].

These chromatin condensation steps are, in reality, much more complex. For example, histones undergo post-translational modifications (PTMs) during their replacement, and all these modifications as well as the proteins linked to the sperm DNA are considered as epigenetic marks, potentially transmissible to the offspring and with an impact on the development of the future embryo [2,9,10,11,12,13,14]. The potential association of residual histones with promoters of specific genes and other genomic regions relevant to embryo development gave rise to the concept that such retained histones may regulate gene expression in the early embryo [2,15,16,17,18]. Furthermore, such histones retained in sperm may carried various PTMs that could participate in regulating gene expression. PTMs include mono-, di-, and trimethylation of lysine residues, acetylation, phosphorylation, ubiquitination, ADP-ribosylation, crotonylation, and others (for a review, see [19,20,21,22,23,24,25,26]).

These PTMs, as epigenetic marks, could also be modified by the environment and have an impact on embryo development (for a review, see [20,21,22,23,24,25,26,27]). As the percentage of sperm-retained histones and the PTMs involved can influence the success of embryo development, it becomes necessary to be able to select the spermatozoa presenting the “best” epigenetic pattern in human assisted reproductive technologies (ART), that is, those which would be linked to the best and safest embryo development. Nowadays, one of the ways of selecting a spermatozoon before injection into the human oocyte is use of high-magnification microscopy with differential interferential contrast (DIC), also called IMSI (intracytoplasmic morphologically selected sperm injection) [28,29,30,31,32]. This technique allows the selection of morphometrically normal spermatozoa without vacuoles and without chromatin granulation, morphological abnormalities related to defects in sperm chromatin condensation [28,29,30,31,33]. The vacuoles were first described as nuclear craters or thumbprints, or DNA-free areas, in connection with no or low condensation of the sperm chromatin [29,31,34,35,36]. Hence, deselecting sperm with chromatin condensation abnormalities could improve pregnancy rates and decrease miscarriage rates [33,37,38,39,40,41,42,43,44,45].

This knowledge of the degree of chromatin condensation in spermatozoa with vacuoles is, however, rather incomplete, since the markers used until now were indirect markers of the presence of retained histones, such as aniline blue or chromomycin A3. Direct markers of these histones or epigenetic marks have not yet been used. To further understand the links between sperm vacuoles and epigenetic marks carried by human spermatozoa, the study presented here aimed at (1) assessing the epigenetic status, through nuclear proteins and PTMs profile, of sperm chromatin at the gamete level according to the presence of sperm-head vacuoles, and (2) examining their spatial distribution in relation to vacuole localization.

## 2. Materials and Methods

### 2.1. Patients

Among couples attending the Assisted Human Reproduction Laboratory of the hospital center of Poissy Saint-Germain-en-Laye from September 2018 to May 2021, ten men previously diagnosed (<3 months) as normozoospermic according to WHO criteria [46], and according to David’s modified classification regarding sperm morphology [47], and ten diagnosed as oligo-astheno-teratozoospermic were included. The etiologies of the couples’ infertility were idiopathic (n = 6), exclusively male (n = 5), exclusively female (n = 4), or both male and female (n = 5). The average age of the patients was 35.2 years. All patients had a normal karyotype. The patients did not present any signs of infection or autoimmune abnormalities. Specifically, no patient had an abnormal round cell concentration. The main sperm characteristics of the 20 patients are summarized in Table 1.

The number of samples (patients) to be included (sample size) was calculated on the basis of the results obtained for protamine 2 labeling (labeling that we had from the existing literature, considered the most significant and clinically informative). By considering:-alpha (the type I error, i.e., the probability of wrongly rejecting H0 and detecting a statistically significant difference when the groups are not actually different) at 0.05 and-beta (the type II error, i.e., the probability of wrongly accepting H0 and not detecting a statistically significant difference when a specified difference between the groups in reality exists) at 0.1, the power of our analyses reached 90% (1-beta: 0.9) with 12 patients included. As we had already included 20 patients, we analyzed the results obtained in these 20 patients, thus exceeding the required number of patients.

The semen analyses were carried out by qualified technicians in our ISO 9001-certified and COFRAC-accredited laboratory. The sperm analyses performed in our laboratory respect the standardization criteria as described in the checklist published by Bjorndahl et al. [48] (Appendix A). According to French legislation, the 20 patients were informed and agreed to participate in this study. The study obtained a favorable opinion from the local ethics committee of the Poissy Saint-Germain-en-Laye hospital, and this was officially confirmed by the Institutional Review Bord (IRB) of the French-speaking andrology society (IRB SALF IRB00012652-19).

### 2.2. Sperm Preparation

Sperm samples were collected by masturbation after 2–7 days of abstinence [46]. Spermatozoa were prepared using migration and centrifugation over a density gradient column (40% to 80%, PureSperm 100, Nidacon, Mölndal, Sweden). A pellet containing motile spermatozoa was reconstituted in a HEPES-buffered medium (Ferticult Flushing medium, Fertipro N.V., Beermen, Belgium).

### 2.3. Sperm Selection under High-Magnification Microscopy (IMSI-like Methodology)

Morphological sperm selection was realized inside a petri dish (WillCo-dish, WillCo Wells BV, Amsterdam, The Netherlands) from a drop (5–10 µL) of sperm fraction obtained after density-gradient selection, diluted in 5 µL of polyvinyl pyrrolidone (10% in Ferticult Flushing medium, FertiPro N.V., Beermen, Belgium).

Motile spermatozoa were observed at a magnification of over ×10,000 at room temperature with an inverted microscope (Eclipse 2000-U, Nikon Optics, Tokyo, Japan, equipped with differential interference contrast (DIC) optics and a ×100 dry objective lens). The data in the literature do not allow us to define the best technique for observing spermatozoa at high magnification [49]. In this study, we used a ×100 dry lens, coupled with ×10 magnification in the eyepieces, and we used a zoom camera allowing ×100 magnification, that is, a total magnification of ×1000 in the eyepieces and between ×1000 and ×10,000 when viewing the camera. Sperm-head vacuoles can have different sizes and distributions. However, vacuoles’ surface size might not correspond to their internal size, so small vacuoles at the cell surface could in fact be larger and deeper internally when observed with three-dimensional microscopy [34]. To avoid such confusion and to guarantee a large internal size for distribution analyses, only morphometrically normal spermatozoa containing a single vacuole occupying more than 15% of head surface were used as vacuolated spermatozoa and compared to morphometrically normal spermatozoa with no vacuoles (Figure 1). This selection technique was made possible by using layers mimicking different vacuole sizes, as previously described [29,50]. For each patient, the same trained and skilled operator quickly observed and classified spermatozoa into four grades, according to the criteria of Vanderzwalmen [51], and selected grade I and grade III spermatozoa (grade I: morphometrically normal motile spermatozoa with a head of normal shape and size, oval, regular, and without vacuoles; grade III: morphometrically normal motile spermatozoa that differ from grade I spermatozoa only by the presence of at least one large vacuole) using fixed, transparent, celluloid outlines of a sperm nucleus and a vacuole occupying more than 15% of the sperm head were used (as first described by Bartoov et al. in 2002 [52] and later by other authors [29,50]). Selection was performed at room temperature to avoid heat-induced vacuolation, as described by Peer et al. in 2007 [53].

The selected spermatozoa were distributed over five 10-well slides in phosphate-buffered saline (PBS) (Biomérieux, France): 20–25 normal spermatozoa in one well and 20–25 vacuolated spermatozoa on the diagonal opposite well. Altogether, more than 4000 spermatozoa (n = 4193 spermatozoa) were assessed, and for each epigenetic mark, more than 800 spermatozoa (more than 20 vacuolated and 20 vacuole-free spermatozoa from the twenty patients) were analyzed at the gamete level. The protocol is summarized in Figure 2. After sperm selection, slides were immediately air-dried, and spermatozoa fixed with cold methanol for 10 min at 4 °C. After fixation, slides were air-dried and conserved at −20 °C until further use.

### 2.4. Selection of Epigenetic Marks of Interest in Male Fertility: Nuclear Proteins and PTMs

A list of all the nuclear proteins and PTMs described in human sperm was established through a preliminary screening of the PubMed database of references and abstracts and PubMed Central (PMC) websites using the following keywords: “(“sperm” OR “male infertility” OR “embryo development”, OR “pregnan*” OR “miscarriage” OR “human embryo”) AND (“chromatin” OR “epigen*” OR “protamine” OR “histone” OR “H3” OR “H4” OR “lysine” OR “post-translationnal modifications”)”. The review returned 137 marks, all summarized in Table 2 [20,21,22,23,24,25,26]. For each of these epigenetic marks, three authors (L.C., M.B., and F.B.) determined their relationship with male infertility, their intra- and inter-individual variability, and their potential relationship with animal or human embryo development. Thus, two nuclear proteins, protamine 2 (P2) and histone 3 (H3), and eight PTMs, H3K4me1, H3K4me2, H3K4me3, H3K9me1, H3K9me2, H3K9me3, H3K27me3, and H4k20me2, were selected and targeted for this study. Such epigenetic marks were chosen for their abundance in sperm and for their potential link with male fertility and/or embryo development.

### 2.5. Fluorescent Immunocytochemistry and Two-Dimensional (2D) Microscopy

The fluorescent immunocytochemistry workflow is detailed in Figure 2. It has been established for this study based on protocols detailed in other studies [45,54,55,56,57,58]. Briefly, conserved slides and dehydrated spermatozoa were first rehydrated with PBS (Biomerieux, France) for 30 min at room temperature (RT). To get access to nuclear protein and PTMs epitopes, sperm chromatin was decondensed with 10 µL of a mix of dithiothreitol 25 mM (DTT, Sigma-Aldrich, D0632-10G, Saint-Louis, MO, USA), Triton-X100 0.2% (Sigma-Aldrich, X-100), and Heparin Choay^®^ 200 IU/mL (SANOFI, 529623) diluted in PBS. Time decondensation was determined for each well by observation with phase-contrast microscopy (Optiphot microscope, Nikon, Japan) coupled to 40X lens (Ph3 DL 40/0.65, Nikon, Japan) according to spermatozoa coloration and shape (i.e., when spermatozoa, usually birefringent (bright), became grey and swelled). The decondensation was stopped with a wash of PBS (5 min) (Sarrate and Anton 2009), and spermatozoa were immediately fixed again with 10 µL of methanol 100% at RT until drying. Then, cell membranes were permeabilized with 0.2% Triton-X100 diluted in PBS for 20 min at RT. Permeabilization was stopped with a wash of PBS.

For each pair of epigenetic marks (P2-H3 and PTMs) targeted, a rabbit and mouse antibody were used. Pairings were established according to their relationship with gene expression and according to antibody availability (Table 3). To avoid interference between rabbit and mouse antibodies, incubation was realized one by one (sequential targeting): first the rabbit and then the mouse antibody. For each kind of antibody, unspecific epitopes were saturated with 60 µL of a mix of BSA 3% (BSA, Sigma-Aldrich, A7906-500G) and powdered milk 5% (Régilait, France) diluted in PBS for 1 h at 37 °C. Primary rabbit antibodies were incubated for 2 h at 37 °C, while primary mouse antibodies were incubated overnight at 4 °C. After primary antibody incubation, slides were washed (5 min) three times with PBS. Secondary antibodies against rabbit or mouse primary antibodies, respectively, were incubated for 1 h at 37 °C. After 3 washes (5 min) with PBS, spermatozoa nucleus was counter-colored with 4′,6-diamino-2-phenylindole (DAPI). Fluorescence was preserved with a drop of anti-fading mounting medium (Vectashield, H-100), sealed with a cover slip, and conserved at 4 °C in darkness until observation.

Primary and secondary antibodies were used at 1/500 in BSA 1% diluted in PBS. Primary antibodies are detailed in Table 3. Secondary antibodies were coupled to cyanin 3 (Cy™3 AffiniPure Goat Anti-Rabbit IgG, Jackson ImmunoResearch 111-165-003) for rabbit antibodies, and to Fluorescein isothiocyanate (FITC AffiniPure Goat Anti-Mouse IgG, Jackson ImmunoResearch 115-095-003) for mouse antibodies. Thus, rabbit antibodies were visualized in red, while mouse antibodies were visualized in green.

All the slides were observed with two-dimensional fluorescent microscopy (Axio Imager.Z2, Zeiss, France) using an X100 immersion lens (EC Plan-Neofluar 100X/1.3, Zeiss, France). A minimum of 400 spermatozoa were observed for each mark tested (minimum 20 for each patient). Each spermatozoon was observed and classified as having a positive or negative targeting for the corresponding epigenetic mark. Positivity was considered if a strong fluorescence was observed (for example, see Figure 3A–C,G–I), while negativity was considered if weak or inexistent fluorescence was observed (for example, see Figure 3D–F,J–L), as previously described [59,60]. A percentage of positive spermatozoa was calculated for normal and vacuolated spermatozoa for each patient and compared between the two classes of spermatozoa using a Wilcoxon test for non-parametric paired data. Data are reported as mean +/− standard error of the mean (SEM) and as median (Q1–Q3; Q1 is the median of the lower half of the data and Q3 is the median of the upper half of the data). Comparisons were performed on SigmaPlot 11.0 software, and significance was considered when *p* ≤ 0.05.

### 2.6. Three-Dimensional (3D) Microscopy

Finally, for epigenetic marks that were significantly over-represented in vacuolated spermatozoa, normal and vacuolated spermatozoa were reconstructed in 3D by deconvolution microscopy. Epifluorescence images were first acquired using a Nikon TE2000-E microscope at 100X (numerical aperture [NA] = 1.3, pixel/micron conversion factor = 15 pixels/μm), configured for imaging in transmitted light, DIC, and epifluorescence modes, and capable of simultaneous DIC/epifluorescence observation of each spermatozoon. Deconvolution microscopy allows the visualization of the cellular structures of fixed specimens in three dimensions [61]. This technique was used on spermatozoa for the first time in 2011 [29]. Here, a cross-section was measured every 100 nm. We thus obtain a stack of about thirty optical sections over the total thickness of the spermatozoon. After deconvolution of the image (the time required for analysis is approximately one night per spermatozoon), Imaris software (Imaris 7.4; Bitplane, South Windsor, CT, USA) was used to produce a 3D image from the non-deconvolved image stack and thus allowed us to observe the interior of each sperm cell.

## 3. Results

### 3.1. Sperm Epigenetic Marks and Sperm Morphology under High-Magnification Microscopy

Vacuolated spermatozoa were statistically more frequently labelled with H3 and H3K4me3 than morphologically normal spermatozoa without vacuoles (88.1% ± 2.7 and 78.5% ± 5.2 vs. 74.8% ± 4.8 and 49.1% ± 7.4, respectively; *p* = 0.009 and *p* < 0.001) (Table 4, Figure 4).

Conversely, sperm with a large vacuole were statistically less marked by P2 and H3K27me3 than morphologically normal spermatozoa without vacuoles (50.2% ± 6.2 and 63.9% ± 6.3 vs. 82.1% ± 4.4 and 73.6% ± 5.1, respectively; *p* < 0.001 and *p* = 0.028) (Table 4, Figure 4).

For the other epigenetic marks (H3K4me1, H3K4me2, H3K9me1, H3K9me2, H3K9me3, and H4K20me2), no significant difference was reported between the two types of spermatozoa (Table 4, Figure 4).

### 3.2. Anatomical Relationships between Vacuoles and Epigenetic Marks

#### 3.2.1. 2D Microscopy

When observing the statistically more present epigenetic marks in vacuolated spermatozoa (H3 and H3K4me3) under 2D microscopy, the H3 epigenetic mark was always diffuse and occupied the entire sperm head in all H3-marked vacuolated spermatozoa (n = 354) (Figure 5A–C).

In contrast, H3K4me3 labelling was always localized and was prevalent in certain areas corresponding to DAPI-free zones in all H3K4me3-labelled vacuole spermatozoa (n = 330) (Figure 5D–F).

Morphometrically normal spermatozoa without vacuoles and those negatively labelled with H3 and H3K4me3 are shown in Figure 6.

#### 3.2.2. 3D- Deconvolution Microscopy

Based on 2D observations, it was decided to reconstruct morphometrically normal H3K4me3-unlabelled spermatozoa and vacuolated H3K4me3-labelled spermatozoa in three dimensions to assess the anatomical distribution of H3K4me3 relative to the vacuole (Figure 7 and Figure 8).

Normal spermatozoa without vacuole contained a DAPI-labelled nucleus (nucleus observed in different cross-sections; see Figure 7C,E–G). H3K4me3 labelling was weak (Figure 7D–G).

For vacuolated H3K4me3-labelled sperm, the vacuole was a DAPI-free area (Figure 8B,C), labelled in its entirety by H3K4me3 (Figure 8D–G). This was the case in each of the observed vacuolated and H3K4me3-labelled spermatozoa (n = 423 spermatozoa).

## 4. Discussion

According to our study, morphometrically normal spermatozoa with a vacuole occupying more than 15% of the head surface show a different epigenetic profile than normal, non-vacuolated spermatozoa. Vacuoles were identified as nuclear areas enriched in H3K4me3.

In our study, vacuoles are observed as DAPI-free areas. This had already been reported previously [29,31,34]. These DAPI-free areas were, until now, wrongly described as DNA-free areas [29,34]. Now, according to data presented, vacuoles can be defined as areas enriched in retained histones. Those retained histones are probably associated with DNA. However, the hypothesis of the presence of DNA-free nucleoplasm enriched in H3K4Me3 cannot be excluded. The sperm vacuole is now described as H3K4me3 enriched. It can be hypothesized that DAPI labelling of the nucleus in these nuclear areas of weakly condensed chromatin is too weak to be observed by fluorescence microscopy.

Excessive retention of histone H3 and the presence of H3K4me3 are statistically over-represented, and by contrast, the presence of protamine 2 (P2) and H3K27me3 are statistically under-represented in vacuolated spermatozoa compared to normal spermatozoa with no vacuoles. The nuclei of vacuolated spermatozoa show excess retention of histone H3 and a protamination defect compared to normal spermatozoa. A protamination defect with decreased protamine P2 was recently observed in vacuolated sperm [62]. This protamination defect allows one to define an immature status of the spermatozoon. Indeed, during spermatogenesis, and particularly during the transitional stages from spermatid to spermatozoon status, the majority of histones have to be replaced by protamines. As histone H3 is one of the most abundant histones, a lack of its replacement leads to poor or non-condensed chromatin. Thus, sperm chromatin remains immature and therefore more susceptible to external factors and sperm DNA damage [63,64,65,66]. Indeed, several studies have described vacuolated spermatozoa as exhibiting higher sperm DNA fragmentation rates [35,67,68,69,70,71].

These defects in sperm chromatin protamination and chromatin condensation are also related by the differential rate of histone H3 PTMs. The epigenetic marks H3K4me3 and H3K27me3 are indeed statistically differentially detected according to sperm morphology at high magnification. H3K4me3, overrepresented in vacuolated spermatozoa, is an epigenetic mark associated with gene expression that is described as particularly enriched at transcription start sites. In contrast, H3K27me3, statistically underrepresented in vacuolated spermatozoa, is an epigenetic mark associated with gene silencing [14]. During the spermatid transition, many histone PTMs take place [11,22]. Several studies have described that the H3k4me3 mark is present in spermatids and should disappear during spermiogenesis [11,22,72,73]. Conversely, these studies showed that H3K27me3 was an overrepresented mark in mature spermatozoa compared to spermatids [22,73]. Thus, vacuolated spermatozoa are described here as spermatozoa with a spermatid-like epigenetic profile, whether by protamination defects or chromatin condensation defects, or by post-translational modifications of histone H3 contained in their nucleus. Enrichment of H3K4me3 and lower H3K27me3 levels in vacuolated spermatozoa may also support the idea that silencing of some regions of the genome could be compromised.

Furthermore, the epigenetic profile of vacuolated spermatozoa may have deleterious consequences for embryo development. Histones were indeed described as mostly retained on gene promoters with a high content of unmethylated CpG regions and on regulatory elements, suggesting a role in the transcriptional regulation of these genes and genome organization after fertilization of oocytes [16,74,75]. H3K4me3 and H3K27me3 were described as associated to the bivalent domain on promoters linked to developmental genes (i.e., HOX genes and genes under paternal imprinting) [16,74]. To determine the exact location of certain epigenetic marks, Yamaguchi et al. used nucleoplasmin to remove protamines before ChIP-seq analysis. This allowed them to clearly localize histones in sperm chromatin. They showed that enrichment of H3K4me3 was located in CpG-rich promoters genes [75], as had already been partially shown by other teams [12,14,15,76]. Concerning H3K4me3, Lambrot et al. have recently shown that sperm H3K4me3 marked developmental genes and correlated with embryonic gene expression in humans [77]. Other authors have also shown that certain histone methylations have this same role in embryo development, and support the fact that paternal epigenetic information transmission to the embryo occurs through the homogeneous retention of methylated histone in a sperm cell population [78]. Epigenetic alteration of sperm histone methylation (as H3K4me3) profiles has also been shown to result in changes in embryo gene expression and congenital abnormalities that are transmitted from one generation to the next [79,80,81]. Epigenetic marks are indeed inherently transmissible to offspring, and it is now increasingly recognized that certain epigenetic alterations can be accompanied by pathological epigenetic traits in the offspring [82,83,84,85,86,87]. For the first time, in this study, epigenetic profile assessment was performed at the one-cell scale. Hence, the selection of morphometrically normal spermatozoa without vacuoles and the deselection of spermatozoa with vacuoles (H3 and H3K4me3 enriched) appear to be epigenetically favorable to embryo development and safe offspring.

Finally, it has been widely demonstrated that histones and their PTMs are the epigenetic memory of the spermatozoon [81,88,89,90,91]. Thus, the sperm vacuoles could be witnesses of the patient’s history and the environment the man has been confronted with. Indeed, according to the literature, the effects of diet or exposure to certain toxic substances on the epigenetic marks carried by spermatozoa have been studied [84,92,93,94,95]. Furthermore, H3K4me3 was described in rats on a low-folate diet as being altered in sperm at the level of developmental genes and putative enhancers, leading to an increase in the severity of developmental defects in the offspring [96]. Thus, the over-representation of H3K4me3 in human sperm vacuoles could be caused by the patient’s environment and is likely to be transmitted to the offspring [96,97]. These epigenetic marks, and more broadly these vacuoles, would then constitute an adequate indicator for monitoring a particular drug or environmental exposure in men. This will have to be the subject of larger-scale randomized studies, but a simple morphological study such as high-magnification observation of these living spermatozoa could allow the epigenetic profile of spermatozoa to be assessed at the one-cell scale. This would provide a simple tool for real-time assessment of the epigenetic profile of each spermatozoon.

## 5. Conclusions

The epigenetic profile of human morphometrically normal spermatozoa varies according to its morphology at high magnification and, more precisely, according to the presence or absence of sperm-head vacuoles. Vacuoles are nuclear concavities, containing DNA enriched in H3K4me3. The overexpression of certain epigenetic marks (H3 and H3K4me3) in vacuolated spermatozoa could influence embryo development and be transmitted to the offspring. The selection of normal spermatozoa without vacuoles and the deselection of spermatozoa with vacuoles appear to be epigenetically favorable to embryo development and safe offspring. Finally, high-magnification microscopy could become a simple tool to estimate in real time the sperm epigenetic profile and to monitor the exposure to an environmental factor. Large-scale studies are needed to confirm these results.

## Figures and Tables

**Figure 1 cells-11-01788-f001:**
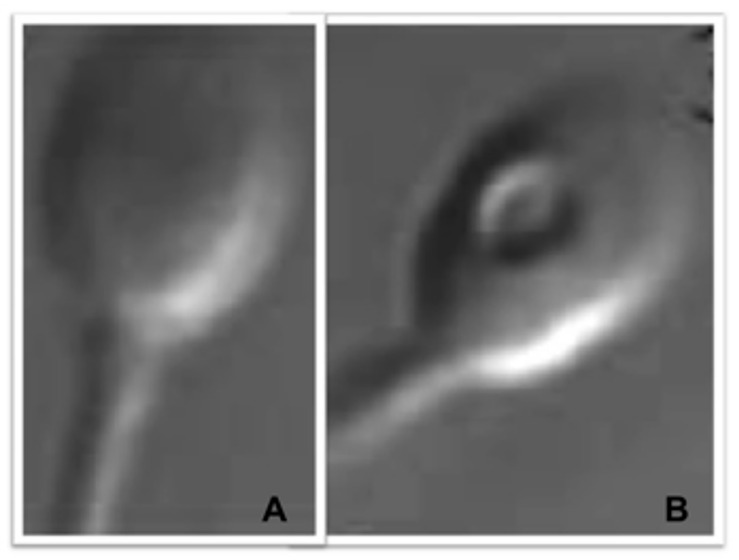
Human spermatozoa observed at high magnification with differential interference contrast (DIC) microscopy (IMSI-like). (**A**) A grade I morphometrically normal spermatozoon with no vacuole. (**B**) A grade III morphometrically normal spermatozoon with one large vacuole.

**Figure 2 cells-11-01788-f002:**
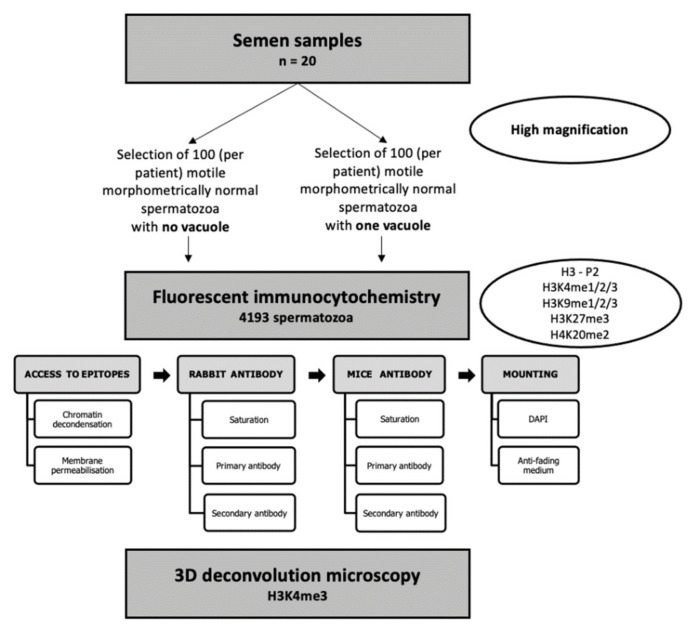
Summary diagram of the experimental workflow.

**Figure 3 cells-11-01788-f003:**
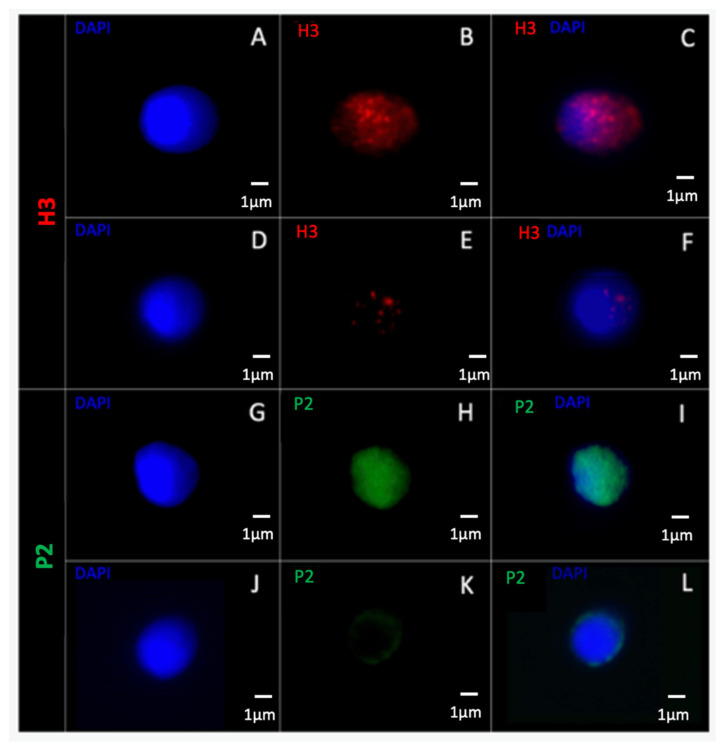
Spermatozoa observed under a two-dimensional fluorescence microscope. A spermatozoon observed with (**A**) DAPI labelling (blue), (**B**) H3-Cy3 positive labelling (red), and (**C**) DAPI/H3-Cy3 merge. Another spermatozoon observed with (**D**) DAPI labelling, (**E**) H3-Cy3 negative labelling, and (**F**) DAPI/H3-Cy3 merge. A spermatozoon observed with (**G**) DAPI labelling, (**H**) P2-FITC positive labelling, and (**I**) DAPI/P2-FITC merge. Another spermatozoon observed with (**J**) DAPI labelling, (**K**) P2-FITC negative labelling, and (**L**) DAPI/P2-FITC merge. H3: histone 3; Cy3: cyanin3; P2: protamine 2; FITC: fluorescein isothiocyanate; DAPI: 4′,6-diamino-2-phenylindole.

**Figure 4 cells-11-01788-f004:**
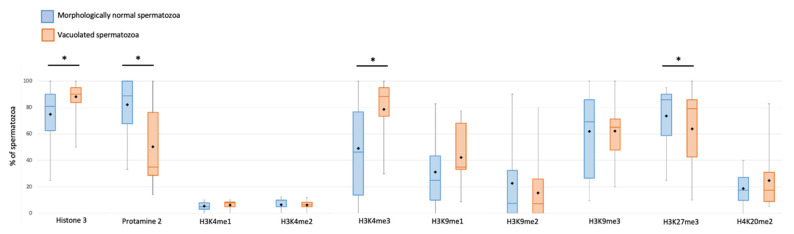
Percentage of non-vacuolated spermatozoa (blue box) and vacuolated spermatozoa (orange box) positively targeted for each of the ten epigenetic marks evaluated. Boxes represent data included in third and first interquartile range (Q3–Q1), horizontal bars represent median, and black squares represent mean. Whiskers represent upper (95%) and lower (5%) confidence intervals. Asterisks indicate significative differences among vacuolated and non-vacuolated spermatozoa (*p* < 0.05).

**Figure 5 cells-11-01788-f005:**
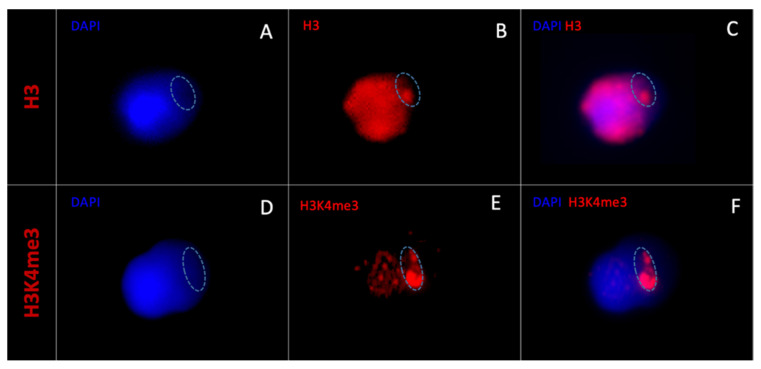
Morphometrically normal spermatozoa with one large vacuole observed with a two-dimensional fluorescence microscope. A vacuolated spermatozoon observed with (**A**) DAPI labelling (blue) and (**B**) histone 3 labelling (red, anti-H3). The vacuole is circled in dotted line. H3 labelling is diffuse and occupies the entire sperm head ((**C**) merged DAPI/H3-Cy3). Another vacuolated spermatozoon observed with (**D**) DAPI labelling and (**E**) H3K4me3 labelling (anti-H3K4me3). The vacuole is circled in dotted line. H3K4me3 labelling is prevalent in certain areas corresponding to DAPI-free zones ((**F**) merged DAPI/H3K4me3-Cy3). DAPI: 4′,6-diamino-2-phenylindole, Cy3: cyanin3.

**Figure 6 cells-11-01788-f006:**
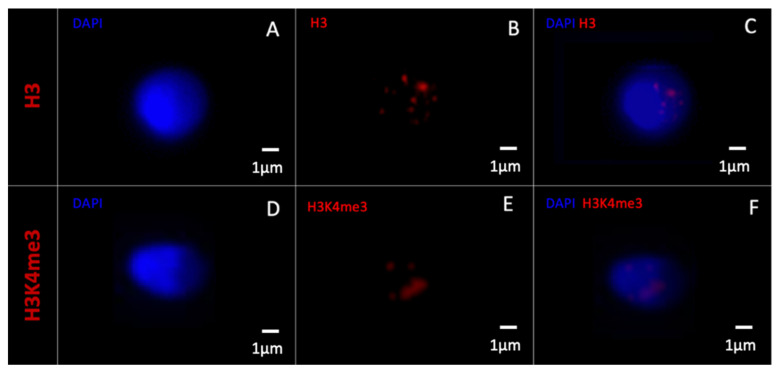
Morphometrically normal spermatozoa without vacuole observed with a two-dimensional fluorescence microscope. A morphometrically normal without vacuole spermatozoon observed with (**A**) DAPI labelling (blue), (**B**) histone 3 labelling (red, anti-H3), and (**C**) merged DAPI/H3-Cy3. Another morphometrically normal without vacuole spermatozoon observed with (**D**) DAPI labelling, (**E**) H3K4me3 labelling (anti-H3K4me3), and (**F**) merged DAPI/H3K4me3-Cy3. DAPI: 4′,6-diamino-2-phenylindole, Cy3: cyanin3.

**Figure 7 cells-11-01788-f007:**
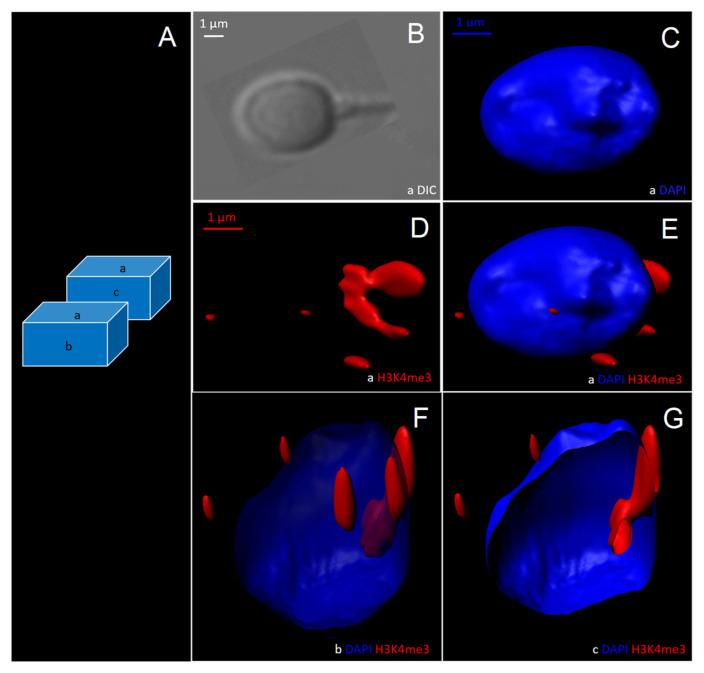
Three-dimensional reconstruction by optical sectional microscopy and deconvolution of a morphometrically normal spermatozoon without vacuole, tagged for H3K4me3. (**A**) Schematic of the upper plane (a), frontal plane (b), and a cross-section (c) of the spermatozoon. (**B**) The spermatozoon visualized with differential interference contrast (DIC). Three-dimensional reconstructed images of the spermatozoon with (**C**) DAPI labelling, (**D**) H3K4me3 labelling, and (**E**–**G**) DAPI/H3K4me3 labelling. DAPI: 4′,6-diamino-2-phenylindole.

**Figure 8 cells-11-01788-f008:**
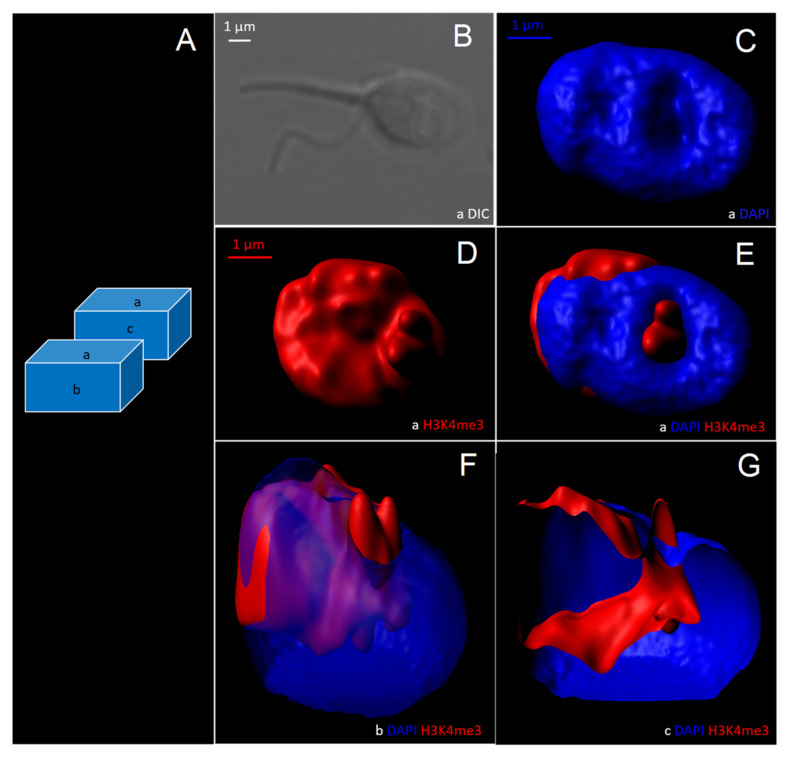
Three-dimensional reconstruction by optical sectional microscopy and deconvolution of a morphometrically normal spermatozoon with one vacuole (occupying more than 15% of head surface), tagged for H3K4me3. (**A**) Schematic of the upper plane (a), frontal plane (b), and a cross-section (c) of the spermatozoon. (**B**) The spermatozoon visualized with differential interference contrast (DIC). Three-dimensional reconstructed images of the spermatozoon with (**C**) DAPI labelling, (**D**) H3K4me3 labelling, and (**E**–**G**) DAPI/H3K4me3 labelling. DAPI: 4′,6-diamino-2-phenylindole.

**Table 1 cells-11-01788-t001:** Main sperm characteristics of the included patients.

Patients	Age	Etiology	Abstinence Time (Days)	Semen Volume (mL)	Sperm Concentration (10^6^/mL)	Total Sperm Count (10^6^/ Ejaculate)	Sperm Vitality (%)	Sperm Total Motility (%)	Sperm Progressive Motility (%)	Sperm Typical Forms ^1^ (%)
1	34	Exclusively female	2	3.2	34.3	109.8	78	60	40	25
2	42	Idiopathic	2	4.3	21.6	92.9	83	45	35	31
3	32	Exclusively female	3	2.6	42.7	111.0	65	50	40	24
4	38	Idiopathic	2	2.8	57.2	160.2	71	60	45	36
5	29	Idiopathic	4	5.9	19.9	117.4	81	60	50	39
6	26	Exclusively female	3	3.3	33.2	109.6	82	50	40	33
7	44	Idiopathic	5	4.1	48.3	198.0	76	60	45	27
8	48	Idiopathic	4	2.3	29.0	66.7	79	60	55	29
9	31	Idiopathic	3	2.6	28.3	73.6	64	45	35	32
10	32	Exclusively female	4	2.7	46.1	124.5	69	50	35	25
11	28	Exclusively male	2	6.4	2.2	14.1	64	30	20	13
12	34	Combined male and female	5	1.9	8.6	16.3	76	50	30	21
13	26	Exclusively male	2	5.8	1.9	11.0	45	20	15	5
14	47	Exclusively male	2	3.1	5.7	17.7	77	40	30	12
15	39	Combined male and female	3	4.2	6.3	26.5	71	40	30	17
16	44	Combined male and female	3	1.7	7.2	12.2	48	20	10	2
17	32	Combined male and female	2	2.5	10.3	25.8	52	30	15	13
18	38	Exclusively male	4	2.1	11.9	25.0	56	30	20	18
19	27	Exclusively male	2	3.0	9.4	28.2	82	50	30	20
20	33	Combined male and female	4	3.2	10.1	32.3	64	40	30	8

^1^ The classification used for the measurement of typical forms is the modified David classification [47].

**Table 2 cells-11-01788-t002:** Nuclear proteins and epigenetics marks of histones described in human spermatozoa (n = 137).

Nuclear Proteins (n = 7)	
Protamine (2)	protamine 1, **protamine 2**
Core histones (5)	H1, H2A, H2B, **H3**, H4
Protamine epigenetic marks (n = 3)	
P1 (3)	ph1, ph27ac1, ph3
P2 (0)	
Histone variants (n = 23)	
H1 (3)	H1.4, H1.t, H1.t2
H2A (11)	H2A.1a, H2A.2b, H2A.2c, H2A.3, H2A.J, H2A.V, H2A.X, H2A-bbd 2/3, H2A.Z, macroH2A.1, macroH2A.2
H2B (6)	tH2B., H2B.1b, H2B.1c/e/f, H2B.1d, H2B.1 l, H2B.2f
H3 (3)	H3, H3.3, H3.1t
H4 (0)	
Histones PTMs (n = 104)	
H1 (4)	K43ac, R50me1, K62me1, K63ac
H1t (13)	K112me1, K113ac, K122ac, K124me1, K170ac, K173me1, K183ac, K183me3, R185me1, S180ph, S187ph, K188me1, K190ac
H2A (3)	K5ac, R11me2, R29me1
H2AV/Z (21)	K4ac, K4me2/3, K7ac, K7cr, K7me1/2/3, S9ph, K11cr, K11me1/2/3, K13cr, K13me2/3, K15ac, R19me1, K27ac, K37cr, K37me1
H2B (2)	K16me1, K20me3
tH2B (15)	K6ac, T9ph, K12ac, K12me1/3, K13me1/3, K16me1/3, K17me1, K28ac, K29ac, R30me2, K86ac, R87me1
H3 (31)	T3ph, **K4me1/2/3**, K9ac, **K9me1/2/3**, K14ac, K18ac, K18me1, K23ac, K23me1/3, R26me1/2 **K27**me1/2/**3**, K36cr, K36me1/2/3, K37ac, K37me2/3, R53me1, K56ac, K79me1/2, K12Ox
H4 (15)	S1ph, R3me1, K5ac, K8ac, K12ac, K16ac, K9ac, **K20me1/2/3**, K31ac, R35me1, M84Ox, K91ac, R92me1

Bolded epigenetic marks correspond to those studied in this manuscript (n = 10). Abbreviations: PTMs = post-translational histone modifications. Modifications: me = methylation, ph = phosphorylation, ac = acetylation, cr = crotonylation, Ox = oxidation. Residues: K = lysine, S = serine, R = arginine, T = threonine, M = methionine.

**Table 3 cells-11-01788-t003:** Pairing of epigenetic marks and primary antibodies used.

Pairing	Rabbit Antibody Cy3 (Red)	Mouse Antibody FITC (Green)
1	Pan H3 Millipore 06-755	Protamine 2 Briar Patch Biosciences Hup-2B
2	H3K27me3 Millipore 07-449	H3K9me2 Active motif 39683 clone MABI 0307
3	H3K9me1 Millipore ABE101	H3K4me2 Millipore 05-1338 clone CMA303
4	H3K4me3 Abcam ab8580	H4K20me2 GeneTex GTX630545 clone GT1851
5	H3K9me3 Active motif 39765	H3K4me1 GeneTex GTX50902 clone MABI0302

For each pair of epigenetic marks targeted, rabbit and mouse antibodies were used. Antibodies marked in red are antibodies coupled to Cyanin 3 (Cy3) and antibodies marked in green are antibodies coupled to Fluorescein isothiocyanate (FITC).

**Table 4 cells-11-01788-t004:** Mean and median values of H3, P2, and PTMs observed in the 20 patients according to sperm morphology.

	Non-Vacuolated Sperm (%)			Vacuolated Sperm (%)			
Nuclear Proteins and PTMs	Mean ± SEM	Median	Q3–Q1	Mean ± SEM	Median	Q3–Q1	*p*-Value
**H3**	74.8 ± 4.8	80.9	62.5–90	88.1 ± 2.7	90	83.7–95	**0.009 ***
**P2**	82.1 ± 4.4	88.8	67.8–100	50.2 ± 6.2	35	28.8–76.3	**<0.001 ***
H3K4me1	5.4 ± 0.8	5.0	3.1–8	6.2 ± 0.8	8	5–8.5	0.325
H3K4me2	6.4 ± 1.0	5.0	5.0–10	6.3 ± 0.9	6.5	5– 8.3	0.71
**H3K4me3**	49.1 ± 7.4	46.3	13.8–76.7	78.5 ± 5.2	88.2	73.3–95	**<0.001 ***
H3K9me1	31.1 ± 6.3	25	10–43.5	42.3 ± 4.8	35	33.3–68.2	0.165
H3K9me2	22.6 ± 6.5	7.5	0.0–32.5	15.4 ± 4.7	73	0.0–25.9	0.13
H3K9me3	61.8 ± 6.9	69.1	26.7–85.7	62.2 ± 4.9	65	47.7–71.2	0.911
**H3K27me3**	73.6 ± 5.1	85.7	58.8–90	63.9 ± 6.3	79.2	42.5–85.9	**0.028 ***
H4K20me2	18.7 ± 2.6	17.4	9.9–27.3	24.7 ± 4.5	17.4	9.1–31	0.121

Values are represented as mean ± standard error to mean (SEM), median, interquartile ranges (Q1–Q3), and estimators (Wilcoxon statistic and *p*-value). Significance is considered when *p* ≤ 0.05 and is indicated by an asterisk *. H3: histone 3, P2: protamine 2, PTMs: post-translational histone modifications.

## Data Availability

The datasets generated and analyzed during the current study are not publicly available but are available from the corresponding author on reasonable request.

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
