# Peer review of "Human Sperm Morphology as a Marker of Its Nuclear Quality and Epigenetic Pattern"

_cells, 2022, doi:10.3390/cells11111788_

Round 1

Reviewer 1 Report

The study systemically evaluated the relationships between sperm carrying morphological abnormalities, nuclear vacuoles in particular, and epigenetic markers retained in sperm.  The results show that histone H3 and H3K4me3 are unusually enriched in vacuolated sperm, whereas protamine P2 and H3K27me2 are reduced.  The presented data is in large part convincing and supportive to the conclusion, which bears both mechanistic and clinical significance.

There are several caveats that presented in the manuscript that may be modified and improved by the authors, including:

1. The conclusion of DAPI free area contains enriched H3K4me3 suggests “DNA associated with retained histones” (lines 324-325) may be drawn more carefully, as the possibility that DNA-free nucleoplasm within nucleus could retain methylated H3 without associating with DNA, for example, not degraded, also exists. Is there any evidence showing the vacuoles do contain DNA positively?

2. Some of the results should be presented in more details, for example, Section 3.2.1. Staining images for the control group (normal sperm) should be provided for Figure 5.  Relevant to the immunostaining results, it may be clearer if the antigens and samples used are indicated in Figure 3 as well.  The results in Figure 6 of Section 3.2.2. should be introduced in more details in the main text.

3. It will be interesting if immuno-staining results of the other epigenetic markers, i.e. Protamine P2 and H3K27me3, can also be presented for normal and abnormal samples.

4. Provide scale bars for all presented images if possible.

5. Some spelling and grammar errors in the text.

Author Response

Dear Editor, dear reviewer,

We would like to thank the two reviewers for the quality of their comments.

All of their comments helped to improve the manuscript.

We hope you will be satisfied with our responses.

We detail our responses in the comments. They are also highlighted in the manuscript.

Sincerely,

Reviewer 2 Report

The authors conducted an excellent study on the nuclear proteins and post-translational histone modifications in morphologically normal spermatozoa with and without nuclear vacuoles. The results obtained have great applied interest for the sperm selection in ICSI programs.

This reviewer has only some concerns related to the number of patients included in the study, and the number of spermatozoa analysed per patient. Therefore, in the Material and Methods section authors must justify that this sample size is enough to obtain robust results from a statistical point of view. Otherwise, they must clearly indicate that this is a preliminary study.

Moreover, considering that their study does not include any approach to determine the relation between the presence of nuclear vacuoles and its effects on the embryo development and offspring, author must eliminate the expression “implications for embryo development and offspring” from the title (line 3).

The Introduction contains some misconceptions:

  • Line 38. Please, note that the sperm maturation occurs in the epididymis, and that the spermiogenesis is the latest stage of the spermiogenesis, which result in the spermatid differentiation into spermatozoa. Therefore, authors must revise this sentence and provide an accurate message.
  • Line 46. Please, refer to “epididymal maturation” instead of “epididymal transit phase”.

Other minor changes that authors must address are detailed below:

  • Line 173. “Bolded epigenetic marks correspond to those studied in this manuscript (n=10). PTMs: post-translational histone modifications.” must be placed in the foot of Table 2 instead of table title.
  • The abbreviations of Table 2 must be placed in the table foot instead of being a part of the table information.
  • Lines 212-213. “For each pair of epigenetic mark targeted, a rabbit and mice antibodies were used.” must be placed in the foot of Table 3 instead of table title.
  • Lines 257 and 260. Please, refer to “morphologically normal spermatozoa without vacuoles” instead of “normal spermatozoa without vacuoles”. Please, also correct Figure 4 accordingly.
  • Lines 266-269. “Values are represented as mean ± standard error to mean (SEM), median, interquartile ranges (Q1 – Q3) and estimators (Wilcoxon statistic and P- value). Significance is considered when P ≤ 0.05 and is indicated by an asterisk*. H3: Histone 3, P2: Protamine 2, PTMs: post-translational histone modifications.” Again, these lines must be placed in the foot of Table 4 instead of table title.
  • The text contains several typographical errors that must be corrected.

Author Response

(The authors gave the same response as above.)

Round 2

Reviewer 2 Report

After reviewing the revision version of the manuscript, I consider that it is acceptable for publication in the present form.